# Greedy Modality Selection via Approximate Submodular Maximization

**Runxiang Cheng**[*1]     **Gargi Balasubramaniam**[*1]     **Yifei He**[*1]     **Yao-Hung Hubert Tsai**[2]     **Han Zhao**[1]

[1]University of Illinois Urbana-Champaign, Illinois, USA
[2]Carnegie Mellon University, Pennsylvania, USA

## Abstract

Multimodal learning considers learning from multi-modality data, aiming to fuse heterogeneous sources of information. However, it is not always feasible to leverage all available modalities due to memory constraints. Further, training on all the modalities may be inefficient when redundant information exists within data, such as different subsets of modalities providing similar performance. In light of these challenges, we study *modality selection*, intending to efficiently select the most informative and complementary modalities under certain computational constraints. We formulate a theoretical framework for optimizing modality selection in multimodal learning and introduce a utility measure to quantify the benefit of selecting a modality. For this optimization problem, we present efficient algorithms when the utility measure exhibits monotonicity and approximate submodularity. We also connect the utility measure with existing Shapley-value-based feature importance scores. Last, we demonstrate the efficacy of our algorithm on synthetic (Patch-MNIST) and real-world (PEMS-SF, CMU-MOSI) datasets.

## 1 INTRODUCTION

Multimodal learning considers learning with data from multiple modalities (e.g., images, text, speech, etc) to improve generalization of the learned models by using complementary information from different modalities.[1] In many real-world applications, multimodal learning has shown superior performance [Bapna et al., 2022, Wu et al., 2021], and has demonstrated a stronger capability over learning from a single modality. The advantages of multimodal learning

---

[*]Equal contribution.
[1]We use the terms modality/view interchangably.

have also been studied from a theoretical standpoint. Prior work showed that learning with more modalities achieves a smaller population risk [Huang et al., 2021], or utilizing cross-modal information can provably improve prediction in multiview learning [Zhang et al., 2019] or semi-supervised learning [Sun et al., 2020]. With the recent advances in training large-scale neural network models from multiple modalities [Devlin et al., 2018, Brown et al., 2020], one emerging challenge lies in the *modality selection* problem.

From the modeling perspective, it might be tempting to use all the modalities available. However, it is inefficient or even infeasible to learn from all modalities as the total number of input modalities increases. A modality often consists of high-dimensional data. And model complexity can scale linearly or exponentially with the number of input modalities [Zadeh et al., 2017, Liu et al., 2018], resulting in large consumption of computational and energy resources. The marginal benefit from the new modalities may also decrease as more modalities have been included. In some cases, learning from fewer modalities is sufficient to achieve the desirable outcome, due to the potential overlap in the information provided by these modalities. Furthermore, proactively selecting the modalities most informative towards prediction reduces the cost of collecting the inferior ones. For example, in sensor placement problems where each sensor can be treated as a modality, finding the optimal subset of sensors for a learning objective (e.g., temperature or traffic prediction) eliminates the cost of maintaining extra sensors [Krause et al., 2011].

In light of the aforementioned challenges, in this paper, we study the optimization problem of modality selection in multimodal learning: given a set of input modalities and a fixed budget on the number of selected modalities, how to select a subset that optimizes prediction performance? Note that in general this problem is of combinatorial nature, since one may have to enumerate all the potential subsets of modalities in order to find the best one. Hence, without further assumptions on the structure of the underlying prediction problem, it is intractable to solve this modality selection problem exactly and efficiently.

*Accepted for the 38th Conference on Uncertainty in Artificial Intelligence* (UAI 2022).

To approach these challenges, we propose a utility function that conveniently quantifies the benefit of any set of modalities towards prediction in most typical learning settings. We then identify a proper assumption that is suitable for multimodal/multiview learning, which allows us to develop efficient approximate algorithms for modality selection. We assume that the input modalities are approximately conditionally independent given the target. Since the strength of conditional independence is now parameterized, our results are generalizable to problems on multimodal data with different levels of conditional independence.

We show that our definition of utility for a modality naturally manifests as the Shannon mutual information between the modality and the prediction target, in the setting of binary classification with cross-entropy loss. Under approximate conditional independence, mutual information is monotone and approximately submodular. These properties intrinsically describe the empirical advantages of learning with more modalities, and allow us to formulate modality selection as a submodular optimization problem. In this context, we can have efficient selection algorithms with provable performance guarantee on the selected subset. For example, we show a performance guarantee of the greedy maximization algorithm from Nemhauser et al. [1978] under approximate submodularity. Further, we connect modality selection to marginal-contribution-based feature importance scores in feature selection. We examine the Shapley value and Marginal Contribution Feature Importance (MCI) [Catav et al., 2021] for ranking modality importance. We show that these scores, although are originally intractable, can be solved efficiently under assumptions in the context of modality selection. Lastly, we evaluate our theoretical results on three classification datasets. The experiment results confirm both the utility and the diversity of the selected modalities. To summarize, we contributes the following in this paper:

- Propose a general measure of modality utility, and identify a proper assumption that is suitable for multimodal learning and helpful for developing efficient approximate algorithms for modality selection.

- Demonstrate algorithm with performance guarantee on the selected modalities for prediction theoretically and empirically in classification problems with cross-entropy loss.

- Establish theoretical connections between modality selection and feature importance scores, i.e., Shapley value and Marginal Contribution Feature Importance.

## 2 PRELIMINARIES

In this section, we first describe our notation and problem setup, and we then provide a brief introduction to submodular function maximization and feature importance scores.

### 2.1 NOTATION AND SETUP

We use $X$ and $Y$ to denote the random variables that take values in input space $\mathcal{X}$ and output space $\mathcal{Y}$, respectively. The instantiation of $X$ and $Y$ is denoted by $x$ and $y$. We use $\mathcal{H}$ to denote the hypothesis class of predictors from input to output space, and $\hat{Y}$ to denote the predicted variable. Let $\mathcal{X}$ be multimodal, i.e., $\mathcal{X} = \mathcal{X}_1 \times ... \times \mathcal{X}_k$. Each $\mathcal{X}_i$ is the input from the $i$-th modality. We use $X_i$ to denote the random variable that takes value in $\mathcal{X}_i$, and $V$ to denote the full set of all input modalities, i.e., $V = \{X_1, ..., X_k\}$. Throughout the paper, we often use $S$ and $S'$ to denote arbitrary subsets of $V$. Lastly, we use $I(\cdot, \cdot)$ to mean the Shannon mutual information, $H(\cdot)$ for entropy, $\ell_{ce}(Y, \hat{Y})$ for the cross-entropy loss $\mathbb{1}(Y = 1) \log \hat{Y} + \mathbb{1}(Y = 0) \log(1 - \hat{Y})$, and $\ell_{01}(Y, \hat{Y})$ for zero-one loss $\mathbb{1}(Y \neq \hat{Y})$.

For the simplicity of discussion, we primarily focus on the setting of binary classification with cross-entropy loss[2]. In this setting, a subset of input modalities $S \subseteq V$ and output $Y \in \{0, 1\}$ are observed. The predictor aims to make prediction $\hat{Y} \in [0, 1]$ which minimizes the cross-entropy loss between $Y$ and $\hat{Y}$. The goal of modality selection is to select the subset of input modalities to this loss minimization goal under certain constraints. Our results rely on the following assumption to hold.

**Assumption 2.1** ($\epsilon$-Approximate Conditional Independence). *There exists a positive constant $\epsilon \geq 0$ such that, $\forall S, S' \subseteq V, S \cap S' = \emptyset$, we have $I(S; S' \mid Y) \leq \epsilon$.*

Note that when $\epsilon = 0$, Assumption 2.1 reduces to strict conditional independence between disjoint modalities given the target variable. In fact, this is a common assumption used in prior work in multimodal learning [White et al., 2012, Wu and Goodman, 2018, Sun et al., 2020]. In practice, however, strict conditional independence is often difficult to be satisfied. Thus, we use a more general assumption above, in which input modalities are approximately conditionally independent. In this assumption, the strength of the conditional independence relationship is controlled by a positive constant $\epsilon$, which is the upper bound of the conditional mutual information between modalities given the target.

**Connection to feature selection.** It is worth mentioning that modality selection shares a natural correspondence to the problem of feature selection. Without loss of generality, a modality could be considered as a group of features; theoretically, the group could even contain a single feature in some settings. But a distinction between these two problems lies in the feasibility of conditional independence. In multimodal learning where input data is often heterogeneous, the (approximate) conditional independence assumption is

---

[2]We choose binary class setting for ease of exposition, our general proofs and results directly extend to multi-class setting. We have only used the binary case to derive the conditional entropy (supplementary material), and to further showcase Corollary 3.1

more likely to hold among input modalities. Whereas in the feature level, such an assumption is quite restrictive [Zhang et al., 2012], as it boils down to asking the data to approximately satisfy the Naive Bayes assumption.

## 2.2 SUBMODULAR OPTIMIZATION

Submodularity is a property of set functions that has many theoretical implications and applications in computer science. A definition of submodularity is as follows, where $2^V$ denotes the power set of $V$, and the set function $f$ assigns each subset $S \subseteq V$ to a value $f(S) \in \mathbb{R}$.

**Definition 2.1** (Nemhauser et al. [1978])**.** *Given a finite set $V$, a function $f : 2^V \to \mathbb{R}$ is submodular if for any $A \subseteq B \subseteq V$, and $e \in V \setminus B$, we have $f(A \cup \{e\}) - f(A) \geq f(B \cup \{e\}) - f(B)$.*

In other words, adding new elements to a larger set does not yield larger marginal benefit comparing to adding new elements to its subset. One common type of optimization on submodular function is submodular function maximization with cardinality constraints. It asks to find a subset $S \subseteq V$ that maximizes $f(S)$ subject to $|S| \leq q$. Finding the optimal solution to this problem is $\mathcal{NP}$-hard. However, Nemhauser et al. [1978] propose that a greedy maximization algorithm can provide a solution with approximate guarantee to the optimal solution in polynomial time. We provide the pseudocode of this greedy algorithm below.

---

**Algorithm 1:** Greedy Maximization

**Data:** Full set $V = \{X_1, ..., X_k\}$, constraint $q \in \mathbb{Z}^+$.
**Input:** $f : 2^V \to \mathbb{R}$, and $p \in \mathbb{Z}^+$, where $p \leq q \leq |V|$
**Output:** Subset $S_p$
$S_0 = \emptyset$
**for** $i = 0, 1, ..., p - 1$ **do**
$\quad X^i = \arg\max_{X_j \in V \setminus S_i} (f(S_i \cup \{X_j\}) - f(S_i))$
$\quad S_{i+1} = S_i \cup \{X^i\}$

---

In this algorithm, $V$ is the full set to select elements from, $f$ is the submodular function to be maximized, $p$ is the number of iterations for the algorithm to run, and $q$ is the cardinality constraint. It starts with an empty set $S_0$, and subsequently adds to the current set $S_i$ the element $X^i$ that maximizes the marginal gain $f(S_i \cup \{X_j\}) - f(S_i)$ at each iteration $i$. Algorithm 1 runs in pseudo-polynomial time $\mathcal{O}(p|V|)$, and has an approximation guarantee as follows.

**Theorem 2.1** (Nemhauser et al. [1978])**.** *Let $q \in \mathbb{Z}^+$, $S_p$ be the solution from Algorithm 1 at iteration $p$, and $e$ is the Euler's number, we have:*

$$f(S_p) \geq (1 - e^{-\frac{p}{q}}) \max_{S:|S| \leq q} f(S) \qquad (1)$$

$\max_{S:|S| \leq q} f(S)$ is the optimal value from the optimal subset whose cardinality is at most $q$. If $f$ is monotone, $\arg\max_{S:|S| \leq q} f(S)$ has cardinality exactly $q$. By running Algorithm 1 for exactly $q$ iterations, we obtain a greedily-obtained value that is at least $1 - \frac{1}{e}$ of the optimal value.

## 2.3 FEATURE IMPORTANCE SCORES

The feature importance domain in machine learning studies scoring methods that measure the contribution of individual features. A common setting of these feature importance scoring methods is to treat each feature as a participant in a coalitional game, in which all of them contribute to an overall gain. Then a scoring method assigns each feature a importance score by evaluating their individual contributions. Many notable feature importance scores are adapted from the Shapley value, defined as follows:

**Definition 2.2** (Shapley [1952])**.** *Given a set of all players $F$ in a coalitional game $v : 2^F \to \mathbb{R}$, the Shapley value of player $i$ defined by $v$ is:*

$$\phi_{v,i} = \sum_{S \subseteq F \setminus \{i\}} \frac{|S|!(|F| - |S| - 1)!}{|F|!} (v(S \cup \{i\}) - v(S))$$
$$(2)$$

The Shapley value of a player $i$ is the average of its marginal contribution in game $v$ in each possible player subsets excluding $i$. The game $v$ is a set function that computes the gain of a set of players. Computing the exact Shapley value of a player is $\sharp\mathcal{P}$-hard, and its complexity is exponential to the number of players in the coalitional game – there are $\mathcal{O}(2^{|F|})$ unique subsets, and each subset $S$ could have a unique $\triangle v(i|S)$ value [Roth, 1988, Winter, 2002]. Nonetheless, in certain game settings, there are approximation methods to Shapley value, such as Monte Carlo simulation [Faigle and Kern, 1992]. When Shapley value is adapted to the feature importance domain, each input feature is a player, and $v$ is also called the *evaluation function*. But $v$ is not unique – it can be a prediction model or utility measure; different $v$ may induce different properties to the Shapley value.

We also examine another feature importance score from Catav et al. [2021], known as the Marginal Contribution Feature Importance (MCI). Kumar et al. [2020] has shown that Shapley-value-based feature importance scores [Shapley, 1952, Lundberg and Lee, 2017, Covert et al., 2020] could underestimate the importance of correlated features by assigning them lower scores if these features present together in the full set. In light of this, MCI is proposed to overcome this issue. In Definition 2.3, MCI of a feature $i$ is the maximum marginal contribution in $v$ over all possible feature subsets. The complexity of computing the exact MCI of a feature is also exponential to the number of features.

**Definition 2.3** (Catav et al. [2021]). *Given a set of all features $F$, and a non-decreasing set function $v : 2^F \rightarrow \mathbb{R}$, the MCI of feature $i$ evaluated on $v$ is:*

$$\phi_{v,i}^{mci} = \max_{S \subseteq F}(v(S \cup \{i\}) - v(S)) \qquad (3)$$

# 3 MODALITY SELECTION

This section presents our theoretical results. In Section 3.1, we introduce the utility function to measure the prediction benefit of modalities, and present its subsequent properties. In Section 3.2, we present theoretical guarantees of greedy modality selection via maximizing an approximately submodular function. In Section 3.3, we show computational advantages of feature importance scores (i.e., Shapley value and MCI) in the context of modality selection. Due to space limit, proofs are deferred to supplementary materials.

## 3.1 UTILITY FUNCTION

In order to compare the benefit of different sets of input modalities in multimodal learning, we motivate a general definition of utility function that can quantify the impact of a set of input modalities towards prediction.

**Definition 3.1.** *Let $c$ be some constant in the output space, and $\ell(\cdot, \cdot)$ be a loss function. For a set of input modalities $S \subseteq V$, the utility of $S$ given by the utility function $f_u : 2^V \rightarrow \mathbb{R}$ is defined to be:*

$$f_u(S) \coloneqq \inf_{c \in \mathcal{Y}} \mathbb{E}[\ell(Y, c)] - \inf_{h \in \mathcal{H}} \mathbb{E}[\ell(Y, h(S))] \qquad (4)$$

In other words, the utility of a set of modalities $f_u(S)$ is the reduction of the minimum expected loss in predicting $Y$ by observing $S$ comparing to observing some constant value $c$. The intuition is based on the phenomena that multimodal input tends to reduce prediction loss in practice. Definition 3.1 can be easily interpretable in different loss functions and learning settings. Note that it is also used in feature selection to measure the unversial predictive power of a given feature [Covert et al., 2020]. Under the binary classification setting with cross-entropy loss, $f_u$ is the Shannon mutual information between the output and multimodal input.

**Proposition 3.1.** *Given $Y \in \{0, 1\}$ and $\ell(Y, \hat{Y}) \coloneqq \mathbb{1}(Y = 1) \log \hat{Y} + \mathbb{1}(Y = 0) \log(1 - \hat{Y})$, $f_u(S) = I(S; Y)$.*

The result above is well-known, and has also been proven in [Grünwald and Dawid, 2004, Farnia and Tse, 2016]. We further can show that $I(S; Y)$ is monotonically non-decreasing on the set of input modalities $S$.

**Proposition 3.2.** *$\forall M \subseteq N \subseteq V, I(N; Y) - I(M; Y) = I(N \setminus M; Y \mid M) \geq 0$.*

A combination of Proposition 3.1 and Proposition 3.2 implies that using more modalities as input leads to equivalent or better prediction. It also shows that Definition 3.1 can quantitatively capture the extra prediction benefit from the additional modalities in closed-form (e.g., $I(N \setminus M; Y \mid M)$). And this extra benefit is the most apparent when test loss reaches convergence (e.g., $\inf$). This monotonicity property is also a key indication that Definition 3.1 can intrinsically characterize the advantage of multimodal learning over unimodal learning.

**Comparison to previous results.** Previous work [Amini et al., 2009, Huang et al., 2021] have discovered similar conclusions that more views/modalities will not lead to worse optimal population error in the context of multiview and multimodal learning, respectively. They obtained this observation through analysis to the excess risks of learning from multiple and single modalities, and show that the excess risk of learning from multiple modalities cannot be larger than that of single modality. Instead, our work adopts an information-theoretic characterization, which leads to an easy-to-interpret measure on the benefits of additional modalities. Furthermore, using well-developed entropy estimators, it is relatively straightforward to estimate these measures in practice. As a comparison, excess risks are hard to estimate in practice, since they depend on the Bayes optimal errors, which limits their uses in many applications.

Next, we show that $f_u(S) = I(S; Y)$ is approximately submodular under Assumption 2.1. Previously, Krause and Guestrin [2012] has shown mutual information to be submodular under strict conditional independence. Here we provide a more flexible notion of submodularity for mutual information. There are also other generalizations of submodularity such as weak submodularity [Khanna et al., 2017] or adaptive submodularity [Golovin and Krause, 2011]. Our definition of approximate submodularity is more specific to the case of mutual information and Assumption 2.1.

**Proposition 3.3.** *Under Assumption 2.1, $I(S; Y)$ is $\epsilon$-approximately submodular, i.e., $\forall A \subseteq B \subseteq V, e \in V \setminus B, I(A \cup \{e\}; Y) - I(A; Y) + \epsilon \geq I(B \cup \{e\}; Y) - I(B; Y)$.*

The above proposition states that if conditional mutual information between input modalities given output is below a certain threshold $\epsilon > 0$, then the utilty function $f_u(\cdot) = I(\cdot; Y)$ admits a diminishing gain pattern controlled by $\epsilon$. This diminishing gain pattern is the definition of submodularity (Definition 2.1). When conditional mutual information is zero, input modalities are strictly conditional independent, and $I(\cdot; Y)$ is strictly submodular.

## 3.2 MODALITY SELECTION VIA APPROXIMATE SUBMODULARITY

With Proposition 3.3, we can formulate the problem of modality selection as a submodular function maximization

problem with cardinality constraint, i.e., $\max_{S \subseteq V} I(S; Y)$ subject to $|S| \leq q$. Usually, $q$ is considerably smaller than $|V|$. However, Theorem 2.1 from Nemhauser et al. [1978] is applicable to $I(\cdot; Y)$ only if it is strictly submodular. There the approximation guarantee differs in our case because the strength of submodularity of $I(\cdot; Y)$ is controlled by the upper bound of conditional mutual information under Assumption 2.1. Under the approximate conditional independence assumption, we show the following result.

**Theorem 3.1.** *Under Assumption 2.1, let $q \in \mathbb{Z}^+$, and $S_p$ be the solution from Algorithm 1 at iteration $p$, we have:*

$$I(S_p; Y) \geq (1 - e^{-\frac{p}{q}}) \max_{S:|S| \leq q} I(S; Y) - q\epsilon \quad (5)$$

To summarize, Theorem 3.1 states that any subset of selected modalities produced by Algorithm 1 has an approximation guarantee, in the setting of classification with cross-entropy loss. Since $I(\cdot; Y)$ is monotonically non-decreasing, we can run Algorithm 1 for $p = q$ iterations to get the best possible greedily-obtained value that is at least $1 - \frac{1}{e}$ of the optimal value minus $q\epsilon$. The $q\epsilon$ term characterizes the fact that, if the to-be-optimized function is not always submodular, the upper bound of conditional mutual information $\epsilon$ could cause a larger approximation error as the algorithm runs longer. Nonetheless, when $\epsilon = 0$, our result in Theorem 3.1 reduces to Theorem 2.1.

Using Theorem 3.1, we can further obtain a bound on the minimum of expected cross-entropy loss and expected zero-one loss achieved by the greedily-obtained set. Let us first denote optimal set $\arg\max_{S:|S| \leq q} I(S; Y)$ as $S^*$, then:

**Corollary 3.1.** *Assume conditions in Theorem 3.1 hold, there exists optimal predictor $h^*(S_p) = \Pr(Y \mid S_p)$ such that*

$$\mathbb{E}[\ell_{01}(Y, h^*(S_p))] \leq \mathbb{E}[\ell_{ce}(Y, h^*(S_p))]$$
$$\leq H(Y) - (1 - e^{-\frac{p}{q}})I(S^*; Y) + q\epsilon \quad (6)$$

Corollary 3.1 shows that the minimum of both losses achieved by $\Pr(Y \mid S_p)$ are no more than the uncertainty of the target output minus the lower bound of our greedily-obtained value from Theorem 3.1. We can also upper bound the difference in minimum cross-entropy losses achieved by the greedily-obtained set and the optimal set.

**Corollary 3.2.** *Assume conditions in Theorem 3.1 hold. There exists optimal predictors $h_1^* = \Pr(Y \mid S_p)$, $h_2^* = \Pr(Y \mid S^*)$ such that*

$$\mathbb{E}[\ell_{ce}(Y, h_1^*(S_p))] - \mathbb{E}[\ell_{ce}(Y, h_2^*(S^*))]$$
$$\leq e^{-\frac{p}{q}} I(S^*; Y) + q\epsilon \quad (7)$$

This result expresses a guarantee on the maximum loss difference from the greedily-obtained set versus the optimal set

using optimal predictors. Both bounds from Corollary 3.1 and Corollary 3.2 are paramterized by the duration and constraint $(p, q)$ of Algorithm 1, as well as the approximation error induced by $\epsilon$. As the algorithm attempts to select a larger set of modalities, both bounds become looser.

Overall, under the setting described in Section 2, the (approximate) submodularity of the utility function allows us to have a solution in polynomial time with approximation guarantee for modality selection under cardinality constraint. Under this theoretical formulation, we can directly extend results of other submodular optimization problems to solve modality selection problems with different constraints and objectives [Wolsey, 1982, Krause and Golovin, 2014].

### 3.3 MODALITY IMPORTANCE

We also examine the possibility of adapting feature importance scores to the context of modality selection, by using them to rank individual modalities. Specifically, we consider Shapley value and MCI. We will show that both the computations of the exact Shapley value and MCI of a modality set is efficient, if our utility function is used as the underlying evaluation function. As previously shown, the utility of a modality $f_u(\{X_i\}) = I(X_i; Y)$ in the classification with cross-entropy loss setting. To proceed, we first show the following propositions for $I(X_i; Y)$.

**Proposition 3.4.** *Under Assumption 2.1, $I(S; Y)$ is $\epsilon$-approximately sub-additive for any $S \subseteq V$, i.e., $I(S \cup S'; Y) \leq I(S; Y) + I(S'; Y) + \epsilon$.*

**Shapley value.** In the classic definition (Definition 2.2), the complexity of computing the exact Shapley value of a player is exponential. However, because Definition 2.2 involves a summation of the marginal contribution $I(S \cup \{X_i\}; Y) - I(S; Y)$, we can leverage the sub-additivity to provide an upper bound of the Shapley value $\phi_{I,X_i}$ via a summation of $I(X_i; Y)$s for all possible subsets. Analogously, the super-additivity should provide a lower bound of $\phi_{I,X_i}$ again expressed by $I(X_i; Y)$. Putting two bounds together gives us an efficient approximation of $\phi_{I,X_i}$. Nonetheless, for $I(S; Y)$ to be super-additive, variables in $S$ must be marginally independent. Thus, we further introduced Assumption 3.1 for this setting. Although Assumption 3.1 is seemingly stronger than Assumption 2.1, it will provide great convenience in approximating the Shapley value of a modality efficiently with a better guarantee parameterized by $\epsilon$, as the following shows.

**Assumption 3.1** ($\epsilon$-Approximate Marginal Independence)**.** *There exists a positive constant $\epsilon > 0$ such that, $\forall S, S' \subseteq V, S \cap S' = \emptyset$, we have $I(S; S') \leq \epsilon$.*

**Proposition 3.5.** *Under Assumption 3.1, $I(S; Y)$ is $\epsilon$-approximately super-additive for any $S \subseteq V$, i.e., $I(S \cup S'; Y) \geq I(S; Y) + I(S'; Y) - \epsilon$.*

**Proposition 3.6.** *If conditions in Proposition 3.4 and Proposition 3.5 hold, we have $I(X_i;Y) - \epsilon \leq \phi_{I,X_i} \leq I(X_i;Y) + \epsilon$ for any $X_i \in V$.*

If Proposition 3.4 holds, the Shapley value of any modality $X_i \in V$ will be upper bounded by its own prediction utility plus $\epsilon$, i.e., $\phi_{I,X_i} \leq I(X_i;Y) + \epsilon$. On the other hand, we can further lower bound the Shapley value if Proposition 3.5 also holds, $I(X_i;Y) + \epsilon \leq \phi_{I,X_i}$. In both bounds, $I(S \cup \{X_i\};Y) - I(S;Y)$ becomes $I(X_i;Y)$, and the summation of all fraction factors in fact equals to 1. If both Proposition 3.4 and Proposition 3.5 hold with $\epsilon = 0$, $I(\cdot;Y)$ is additive, in which case, the Shapley value of a modality is exactly its prediction utility, i.e., $\phi_{I,X_i} = I(X_i;Y)$. Furthermore, by the efficiency property of the Shapley value, we must have $I(V;Y) = \sum_{X_i \in V} \phi_{I,X_i}$.

**MCI.** As claimed by Catav et al. [2021], MCI has an extra benefit over Shapley value (Section 2.3). By its definition, solving MCI of a feature requires $\mathcal{O}(2^{|F|})$, where $|F|$ is the total number of features. But if the evaluation function of MCI is submodular, we can efficiently compute the exact MCI. Using Proposition 3.3, we have the following result.

**Proposition 3.7.** *Under Assumption 2.1, $\forall X_i \in V$, we have $I(X_i;Y) \leq \phi_{I,X_i}^{mci} \leq I(X_i;Y) + \epsilon$.*

If $\epsilon = 0$, $I(S;Y)$ will be strictly submodular for any $S \subseteq V$, and the MCI of a modality is exactly its prediction utility, i.e., $\phi_{I,X_i}^{mci} = I(X_i;Y)$. If Proposition 3.4 holds with $\epsilon = 0$, $I(\cdot;Y)$ is sub-additive, then $I(V;Y) \leq \sum_{X_i \in V} I(X_i;Y) = \sum_{X_i \in V} \phi_{I,X_i}^{mci}$. If Proposition 3.5 further holds with $\epsilon = 0$, then $I(\cdot;Y)$ is additive, we can obtain an efficiency property of the MCI in this problem setting, i.e., $I(V;Y) = \sum_{X_i \in V} \phi_{I,X_i}^{mci}$.

**Modality selection via MCI ranking.** In light of these properties, we can consider ranking individual modalities by Shapley value or MCI as an alternative for modality selection besides greedy maximization. The ranking algorithm computes the Shapley value or MCI for all modalities, and returns the top-$q$ modalities with maximum scores w.r.t. a subset size limit $q$. One advantage of this approach is its complexity of $\mathcal{O}(|V|)$, while greedy maximization requires $\mathcal{O}(q|V|)$. As shown above, solving Shapley value efficiently requires additional assumptions to hold (Assumption 3.1), thus MCI ranking would be more preferable.

## 4 EXPERIMENTS

We present empirical evaluation of greedy maximization (Algorithm 1) and MCI ranking on three classification datasets.

**Patch-MNIST.** Patch-MNIST is a semi-synthetic static dataset built upon MNIST [LeCun and Cortes, 1998]. Specifically, we divide each image in the original MNIST into non-overlapping square patches. Each patch location represents a single modality. We construct and experiment on two Patch-MNIST variants, where one variant has 49 patches and each patch is of size $4 \times 4$ square pixel, and another has 9 patches and each patch has the side length of 9 or 10 pixels. Patch-MNIST has ten output classes, 50,000 training images, and 10,000 testing images.

**PEMS-SF.** PEMS-SF is a real-world time-series dataset from UCI ( Dua and Graff [2017]). This dataset represents the traffic occupancy rate of different freeways of the San Francisco bay area. The classification task is to predict the day of the week. Data is obtained from 963 sensors placed across the bay area, where each sensor represents a single modality. Each sensor has a time series with 144 time steps, which we down-sample to 36 via taking the regional means of size-4 windows. Running Algorithm 1 requires $\mathcal{O}(q|V|)$ with $|V| = 963$, and each step requires training a new model. To mitigate extensive run-time, we experiment on 45 out of 963 sensors by filtering sensors in line for the same freeway. There are a total of 440 instances (days), with the train-val-test split being 200, 67, 173 samples.

**CMU-MOSI.** CMU-MOSI is a popular real-world benchmark dataset in affective computing and multimodal learning [Zadeh et al., 2016]. The task is 3-classes sentiment classification (positive, neutral, negative) from 20 visual and 5 acoustic modalities with temporal features. Specifically, CMU-MOSI collects time-series facial action units and phonetic units from short video clips (10-seconds clip sampled at 5Hz rate). Each unit is a modality, and consists of a 50-dimensional feature vector. Training and testing sample size are 1284 and 686 respectively.

**Independence Assumption Validation** We validate the independence conditions (e.g., Assumption 2.1) on all datasets by comparing the mean conditional Mutual Information (MI) and the mean marginal MI of disjoint modalities [Gao et al., 2017]. As shown in Table 1, the conditional MI is smaller than the marginal MI for MNIST and PEMS-SF. Both conditional and marginal MI are small for CMU-MOSI. This implies that modalities should be approximately conditionally independent in these datasets.

Table 1: Mean Marginal/Conditional Mutual Information

| Dataset | Mean Marg. MI | Mean Cond. MI |
|---|---|---|
| **Patch-MNIST** | 2.187 | 0.078 |
| **PEMS-SF** | 0.626 | 0.223 |
| **CMU-MOSI** | 0.064 | 0.069 |

### 4.1 IMPLEMENTATION

We implement greedy maximization based on the pseudo-code in Algorithm 1. We implement MCI ranking by com-

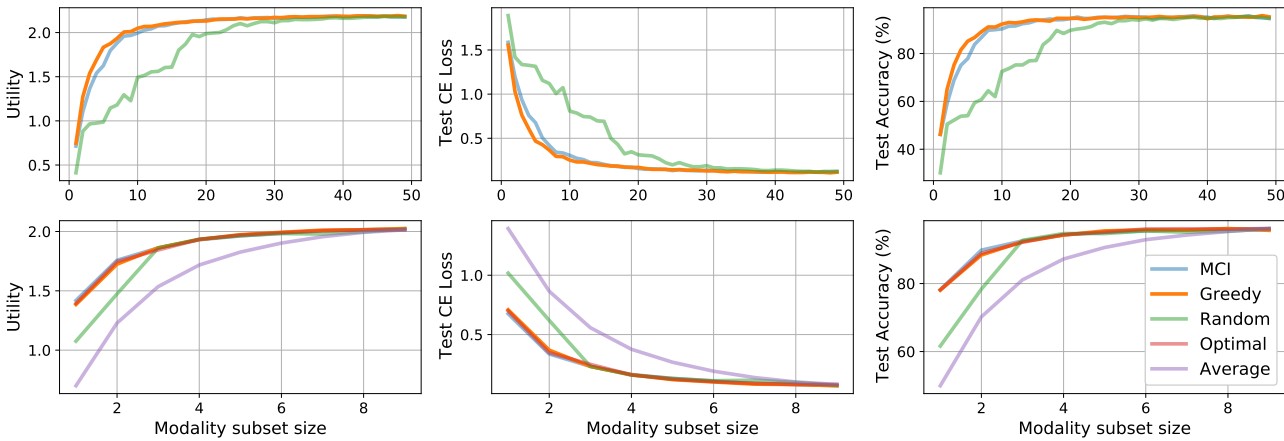

Figure 1: Experiment results for Patch-MNIST with 49 modalities (first row) and with 9 modalities (second row).

puting the MCI for each modality in the full set, and then select the top-ranked modalities with the largest MCIs.

**Utility estimation.** From Proposition 3.1, utility $f_u(S)$ equals $I(S; Y)$, and $I(S; Y) = H(Y) - H(Y \mid S)$. Based on the variational formulation of the conditional entropy as the minimum cross-entropy, we approximate $H(Y \mid S)$ by using the converged training loss on $S$ to predict $Y$ [Farnia and Tse, 2016]. Accordingly, to estimate the marginal gain $I(X_j; Y \mid S_i)$ from Algorithm 1 over high dimensional data, we compute the difference $H(Y \mid S) - H(Y \mid S \cup \{X_j\})$ [McAllester and Stratos, 2020]. To compute MCI of each modality $X_j$, we just need to compute $I(X_j; Y)$, according to Proposition 3.7.

**Modeling.** We now describe models for prediction and utility estimation. For Patch-MNIST, we use a convolutional neural network with one convolutional layer, one max pooling layer and two fully-connected layers with ReLU for both estimation and prediction. The network is trained with Adam optimizer on a learning rate of $1e-3$. For PEMS-SF, we use a 3-layer neural network with ReLU activation and batch normalization for estimation. This is trained with Adam optimizer on a learning rate of $5e-4$. For prediction, we use a recent a time-series classification pipeline [Dempster et al., 2020] for time-series data processing, followed by a linear Ridge Classifier [Löning et al., 2019]. For CMU-MOSI, we experiment with two prediction model types: a linear classifier with Rocket Transformation for time-series (same as the one for PEMS-SF); and a plain 3-layer fully-connected neural network with ReLU activation. On each dataset, the number of training epochs are the same for all evaluated approaches across different modality subset sizes.

### 4.2 EXPERIMENTAL PROCEDURES

In each iteration $i$ of the Algorithm 1 we execute the following: (1) for each candidate modality $X_j$: (a) train two models on $S$ and $S \cup \{X_j\}$ respectively until training losses converge, (b) take the loss difference to be $I(X_j; Y \mid S_i)$; (2) record test loss and accuracy from the model trained on $S_i \cup \{X^i\}$ before the model over-fits; (3) add selected modality $X^i$ to $S_i$ and go to next iteration. We use model parameters before over-fitting for prediction, and parameters after over-fitting for utility estimation.

Step (2) for PEMS-SF and CMU-MOSI are slightly different, in which we record and show the training loss before over-fitting instead of the test loss. This is because PEMS-SF and CMU-MOSI have a much smaller sample size than Patch-MNIST with potentially noisier features, the model likely will not generalize stably. Thus we first examine Theorem 3.1 and MCI ranking on a larger sample set which better represents population and not influenced by the generalization gap. Then we analyze with the test accuracy to accounting the generalization.

For Patch-MNIST with 49 modalities, PEMS-SF and CMU-MOSI, we evaluate Algorithm 1 and MCI ranking against a randomized baseline at each set size. The randomized baseline randomly selects a modality iteratively. For Patch-MNIST with 9 modalities, we further include optimal and average baselines. At each set size $q$, the optimal baseline is the optimal value from all possible subsets of size $q$, and the average baseline is the average. We only implement the optimal baseline for the 9 modalities case because evaluating on all possible subsets for a larger set is expensive.

**Training cost.** At each iteration of Algorithm 1, the marginal utility gain for each candidate modality is evaluated. Since we estimate the conditional mutual information by training a neural network, and we need to evaluate each modality subset at different set sizes, each iteration involves model training. These experiments can be costly for large datasets and models. The training cost at each iteration of Algorithm 1 depends on different utility variants, or mutual information estimation methods in this setting.

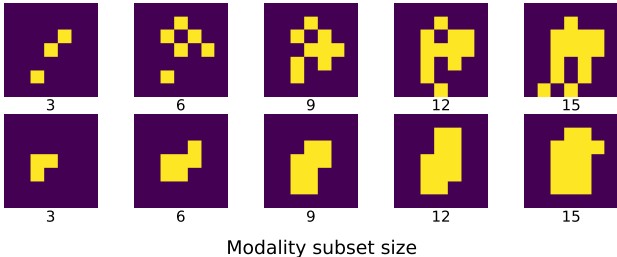

Modality subset size

Figure 2: Modality selection paths of Algorithm 1 (first row) and ranking via MCI (second row) in Patch-MNIST.

## 4.3 RESULTS AND EMPIRICAL ANALYSIS

### 4.3.1 Patch-MNIST

Fig. 1 shows the Patch-MNIST experiment results. In this figure, "Modality subset size" refers to the size of the selected modality set. "Utility" refers to the utility of the selected set. The "Test CE Loss" and "Test Accuracy" refers to the cross-entropy loss and prediction accuracy on test data from the model that is trained on the selected set.

**Utility.** An immediate observation is the high correlation among the utility, test cross-entropy loss and accuracy in both rows. The trend of test accuracy seems identical to the utility, although they mildly differ when the set size exceeds 30. In addition, the utility and test loss is negatively correlated, matching to Definition 3.1. Utility has a larger upper bound than test loss, potentially because the utility is estimated by converging training loss, which is often reduced in greater magnitude than test loss. The utility has a trend of non-decreasing and diminishing gain, which matches the monotonicity and (approximate) submodularity shown in this setting. Adding more modalities is unnecessary if the subset is already large: in the 49-modalities case, accuracy barely improves after 20 modalities selected; but in 9-modalities case, this pattern is less obvious.

**Greedy maximization.** Algorithm 1 beats random selection in both cases. In Fig. 1 (second row), it beats the average by selecting the modality with maximum utility from the start, and overlaps its trajectory with the optimal. In Fig. 1 (first row), Algorithm 1 achieves near-maximum utility with only 7 modalities. These results validate the approximate guarantee from Theorem 3.1. In fact, the guarantee on utility is empirically much better than theoretically proven.

**MCI ranking.** In the 9-modalities case, MCI ranking is as good as greedy maximization and the optimal baseline when the full set has fewer modalities. When more modalities are available for selection (e.g., 49 modalities), Algorithm 1 select a subset that minimizes the loss slightly further than the highest ranked modalities when set size below 15.

**Modality selection path.** We plot the modality selection paths from Algorithm 1 and MCI ranking in Fig. 2. We can

see that MCI selects the modalities that each contain the most information to output – the center regions. Whereas the modalities selected by Algorithm 1 are more diverse, covering different spatial locations of the original image, leading to an advantage in gaining more information collectively.

### 4.3.2 PEMS-SF

Fig. 3 shows our experiment results on PEMS-SF. In Fig. 3, the two leftmost plots show the utility and cross-entropy loss on the training data. The rightmost plot of Fig. 3 shows the moving average of test accuracy instead, because model was not generalized stably under small sample size.

**Utility.** The difference in utility and loss among Algorithm 1, MCI ranking and random baseline are small, and all of them quickly converge to the minimum possible value after selecting only a few modalities. This is potentially because almost each of the modality is sufficient to make training loss small. However greedily selected subsets still has slightly more utility than subsets from MCI ranking and random baseline at every set size. Overall, we still observe the utility is monotone and (approximate) submodular; and Algorithm 1's achieved utility matches Theorem 3.1.

**Generalization.** From the test accuracy plot, we can see a clear advantage from the greedily-obtained set over others when the subset size is small. Meanwhile, MCI ranking is worse than random baseline, which could imply that MCI ranking does not have a robust performance guarantee as Algorithm 1. Other than that, the test accuracy of Algorithm 1 gradually decreases as more modalities are added. This is inline with the over-fitting artifact of greedy feature selection from Blanchet et al. [2008]. However, in the regime of good generalization, greedy maximization should preserve the performance guarantee during testing.

### 4.3.3 CMU-MOSI

The results are alike for both prediction model types mentioned in Section 4.1 for CMU-MOSI. Thus we only use Fig. 4 to show the CMU-MOSI evaluation results from the 3-layer fully-connected neural network. In Fig. 4, the two leftmost plots show the utility and cross-entropy loss on the training data. The rightmost plot of Fig. 4 shows the moving average of test accuracy since the model lacks the capacity to generalize well for this dataset under small sample size.

Overall, many previous observations from other datasets still hold for CMU-MOSI. For example, the utility curve is approximately submodular and monotone as number of selected modalities increases. Modalities selected by Algorithm 1 and MCI ranking outperform randomly selected modalities by having more utility, lower training loss, higher testing accuracy, especially when the number of modalities is still small. On the other hand, potentially due to the sim-

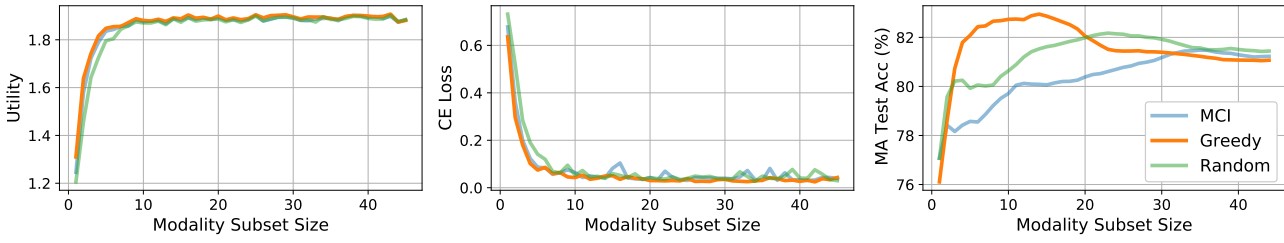

Figure 3: Experiment results for PEMS-SF.

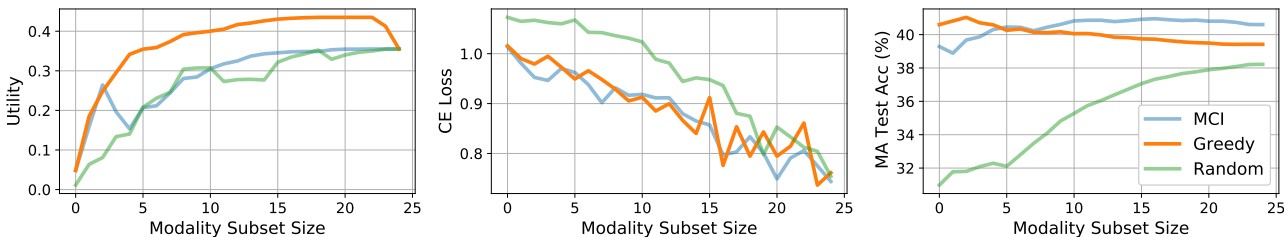

Figure 4: Experiment results for CMU-MOSI.

plicity of the model and noisy features, we are unable to observe an increase of testing accuracy as more modalities are included in Algorithm 1 and MCI ranking.

## 5 RELATED WORK

**Multimodal Learning** Multimodal learning is a vital research area with many applications [Liu et al., 2017, Pittermann et al., 2010, Frantzidis et al., 2010]. Theoretically, Huang et al. [2021] showed that learning with more modalities achieves a smaller population risk, and this marginal benefit towards prediction could be upper bounded. However, the existing measure of marginal benefit [Huang et al., 2021] is hard to understand and cannot be easily estimated, and it does not provide further insight on the emerging modality selection problem.

**Submodular Optimization** Thanks to the benign property of submodularity, many subset selection problems, which are otherwise intractable, now admit efficient approximate solutions [Fujishige, 2005, Iwata, 2008, Krause and Golovin, 2014]. The first study of greedy algorithm over submodular set function dates back to Nemhauser et al. [1978]. Since then, submodular optimization has been widely applied to diverse domains such as machine learning [Wei et al., 2015] , distributed computing, and social network analysis [Zhuang et al., 2013]. A typical type of problem is submodular maximization, which can be subject to a variety of constraints such as cardinality, matroid, or knapsack constraints (Lee et al. [2010], Iyer and Bilmes [2013]). In our case, we extended results from Nemhauser et al. [1978] to the case of approximate submodularity of mutual information in a multimodal learning setting.

**Feature Selection** Feature selection asks to find a feature subset that can speed up learning, improve prediction and provide better interpretability to the data/model [Li et al., 2017, Chandrashekar and Sahin, 2014]. Here we briefly touch related work on feature selection more relevant to our context. Information-theoretic measures such as mutual information have been as a metric for feature selection [Brown et al., 2012, Fleuret, 2004, Chen et al., 2018]. For example, Brown et al. [2012] presents a unified information-theoretic feature selection framework via conditional likelihood maximisation. There are also work on feature selection in regression problems through submodular optimization [Das and Kempe, 2011]. In our context, a distinction between the problems of modality selection and feature selection are the assumptions of the underlying data (Section 2).

## 6 CONCLUSION

In this paper, we formulate a theoretical framework for optimizing modality selection in multimodal learning. In this framework, we propose a general utility function that quantify the impact of a modality towards prediction, and identify proper assumption(s) suitable for multimodal learning. In the case of binary classification under cross-entropy loss, we show the utility function conveniently manifests as Shannon mutual information, and preserves approximate submodularity that allows simple yet efficient modality selection algorithms with approximation guarantee. We also connects modality selection to feature importance scores by showing the computation advantages of using Shaply value and MCI to rank modality importance. Lastly, we evaluated our results on a semi-synthetic dataset Patch-MNIST, and two real-world datasets PEMS-SF and CMU-MOSI.

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
