# OpenReview forum: "Greedy Modality Selection via Approximate Submodular Maximization"
_auai.org/UAI/2022/Conference — UAI 2022 Poster_

### Official Review · Reviewer_Z7aX · 2022-03-20

**Q2(1) Originality/Novelty:** 2
**Q2(2) Significance/Impact:** 2
**Q2(3) Correctness/Technical Quality:** 2
**Q2(6) Clarity Of Writing:** 3
**Q6 Overall Score:** 5
**Q8 Confidence In Your Score:** 4

**Q1 Summary And Contributions:**

The authors propose a new greedy algorithm for selecting features,
apply it to real data. They also extend known theoretical results
that are currently known for the case when features are independent
to a more general case when the features are almost independent in
some reasonable sense.

**Q2 Assessment Of The Paper:**

More detailed information regarding each of these aspects is given below:

**Q2(4) Quality Of Experiments (Optional):**

1: Poor: The experimental evaluation is flawed or the results fail to adequately support the main claims.

**Q2(5) Reproducibility:**

2: Fair: Key resources (e.g., proofs, code, data) are unavailable but key details (e.g., proof sketches, experimental setup) are sufficiently well-described for an expert to confidently reproduce the main results.

**Q3 Main Strengths:**

They extend known theoretical results
that are currently known for the case when features are independent
to a more general case when the features are almost independent in
some reasonable sense.

**Q4 Main Weakness:**

The empirical part is not convincing at all. The authors never
compare their result with previously known greedy algorithms, so it
is not clear whether the new method is better.

Also, the empirical part does not seem to be related to the
theoretical part at all: the authors never check whether the almost
independence condition is satisfied, and for what epsilon.

**Q5 Detailed Comments To The Authors:**

The authors propose a new greedy algorithm for selecting features,
apply it to real data. They also extend known theoretical results
that are currently known for the case when features are independent
to a more general case when the features are almost independent in
some reasonable sense.

The empirical part is not convincing at all. The authors never
compare their result with previously known greedy algorithms, so it
is not clear whether the new method is better.

Also, the empirical part does not seem to be related to the
theoretical part at all: the authors never check whether the almost
independence condition is satisfied, and for what epsilon.

Even if the authors did not compare -- and thus, cannot claim
anything about the quality of their algorithm, this theoretical
part may form a nice short article.

There are, however, many other problems. For example, after
Definition 2.3, the author claim that Shapley value requires
exponential time to compute. This is nonsense, it is like claiming
that since a + b = a + 1 + ... + 1 (b times), the sum a + b
requires exponential time to compute for large b. Actually, there
exist efficient Monte-Carlo algorithms for computing the Shapley
value.

Definition 2.2 should say "defined" instead of "evaluated", it is a
definition after all.

There are also minor English typos: e.g., attempting instead of
tempting on p. 1; line 3 on right-hand side of p. 2 should be "and"
not "or".


**Q7 Justification For Your Score:**

Empirical part is not ready, since the authors never compare the newly proposed method with what was known before. The theoretical part, whole not related to the empirical part, looks promising -- while having some flaws.

**Q9 Complying With Reviewing Instructions:**

1: Yes.

---

### Official Review · Reviewer_xYBK · 2022-04-11

**Q2(1) Originality/Novelty:** 3
**Q2(2) Significance/Impact:** 3
**Q2(3) Correctness/Technical Quality:** 3
**Q2(6) Clarity Of Writing:** 4
**Q6 Overall Score:** 7
**Q8 Confidence In Your Score:** 4

**Q1 Summary And Contributions:**

The inputs space of a machine learning task exhibits multimodality by nature. Ideally, to train a good model, at every step when evaluating the loss function, we should consider all the inputs modalities simultaneously. However, it is not efficient or even infeasible in practice. The authors of this paper developed a greedy heuristic based on mutual information that guarantees the optimal modality selection via submodular maximization.

**Q2 Assessment Of The Paper:**

More detailed information regarding each of these aspects is given below:

**Q2(4) Quality Of Experiments (Optional):**

3: Good: The experimental evaluation is adequate, and the results convincingly support the main claims.

**Q2(5) Reproducibility:**

3: Good: Key resources (e.g., proofs, code, data) are available and key details (e.g., proofs, experimental setup) are sufficiently well-described for competent researchers to confidently reproduce the main results.

**Q3 Main Strengths:**

The motivation of this paper is clear and strong. Modality selection is an important topic and benefits the downstream machine learning tasks in practice. Though submodular optimization is not a novel method in general, people have used it for active learning also, the authors gave theoretical justifications for modality selection, which is counted as the novelty of this paper.

**Q4 Main Weakness:**

If I didn't misread the figures, the improvement regards to other baselines are marginal. Seems to me, that the tasks the authors used are not challenging enough. I will suggest the authors consider more challenging tasks so that the audience could have a clear insight into when the proposed method is better.

**Q5 Detailed Comments To The Authors:**

It is totally fine to assume binary cross-entropy loss to derive the theories. However, could the authors show other loss functions used in the experiment?

**Q7 Justification For Your Score:**

The study presented in this paper is comprehensive enough and easy to follow. This paper is well motivated and the authors gave theoretical justifications. A submodular optimization is a well-studied tool in active learning. The only concern seems the improvement is marginal. However, I am not an expert in feature learning. If authors could append more cases with variants of loss functions in the experiment. That would be more interesting.

**Q9 Complying With Reviewing Instructions:**

1: Yes.

---

### Official Review · Reviewer_6YnH · 2022-04-12

**Q2(1) Originality/Novelty:** 3
**Q2(2) Significance/Impact:** 3
**Q2(3) Correctness/Technical Quality:** 3
**Q2(6) Clarity Of Writing:** 4
**Q6 Overall Score:** 7
**Q8 Confidence In Your Score:** 2

**Q1 Summary And Contributions:**

The presented work proposes a selection method in constrained multi-modality settings. To this end, a utility function for quantification of the usefulness of modalities employed in a greedy search strategy is proposed. This modality utility is further connected to traditional feature selection measurements, such as the Shapley value, allowing for efficiency gains in the modality selection over the greedy approach.

**Q2 Assessment Of The Paper:**

More detailed information regarding each of these aspects is given below:

**Q2(4) Quality Of Experiments (Optional):**

3: Good: The experimental evaluation is adequate, and the results convincingly support the main claims.

**Q2(5) Reproducibility:**

3: Good: Key resources (e.g., proofs, code, data) are available and key details (e.g., proofs, experimental setup) are sufficiently well-described for competent researchers to confidently reproduce the main results.

**Q3 Main Strengths:**

- Solid theoretical analysis of the method proposal incl. consideration of most recent related work
- Compelling link of the utility function to Shapley values / MCI
- Empirical experiments on different types of data (images and time series), overall convincing results

**Q4 Main Weakness:**

- Empirical evaluation involves rather homogeneous, interdependent modalities (contradictory to epsilon-approximate conditional independence, at least for small epsilons)

**Q5 Detailed Comments To The Authors:**

- As e.g. for Patch-MNIST, the patches are embedded in a global spatial structure, which makes them interdependent (at least to some extent). Thus, the modalities cannot be considered as conditionally independent. Albeit your analysis weakens the strict conditional independence by an epsilon approximation, it is still internally relying on a somewhat small epsilon for meaningful / tight bounds. An empirical study with more independent modalities could be insightful.
- Limitations: As described in Section 4.2, each iteration in the greedy algorithm (Algorithm 1) involves a model training. When thinking of larger datasets and models, this is typically not feasible (or at least extremely costly). This should be addressed more explicitly.
- How would your approach generalize to the multi-class setting? It would be nice to have a sort of “outlook”


**Q7 Justification For Your Score:**

This work can certainly be considered as a valuable contribution, where solid theoretical and empirical results support the method proposal. Also, this work connects to previous measurements in the domain of feature selection, which I appreciate.

**Q9 Complying With Reviewing Instructions:**

1: Yes.

---

### Official Review · Reviewer_PQSj · 2022-04-13

**Q2(1) Originality/Novelty:** 2
**Q2(2) Significance/Impact:** 3
**Q2(3) Correctness/Technical Quality:** 3
**Q2(6) Clarity Of Writing:** 3
**Q6 Overall Score:** 5
**Q8 Confidence In Your Score:** 3

**Q1 Summary And Contributions:**

The motivation of the paper is to study modality selection, intending to efficiently select the most informative modalities under certain computational constraints. The contrbutions includes:
(1) The authors formulate a theoretical framework for optimizing modality selection in multimodal learning.
(2) They introduce a general utility function that measures the benefit of selecting a modality.
(3) They establish theoretical connections between modality selection and feature importance scores.

**Q2 Assessment Of The Paper:**

More detailed information regarding each of these aspects is given below:

**Q2(4) Quality Of Experiments (Optional):**

2: Fair: The experimental evaluation is weak: important baselines are missing, or the results do not adequately support the main claims.

**Q2(5) Reproducibility:**

3: Good: Key resources (e.g., proofs, code, data) are available and key details (e.g., proofs, experimental setup) are sufficiently well-described for competent researchers to confidently reproduce the main results.

**Q3 Main Strengths:**

On the whole, the paper has fluent description and clear topic. The paper formulates a good theoretical framework for optimizing modality selection in multimodal learning and the experiments performed in the article validate the proposed method.

**Q4 Main Weakness:**

The topic of the paper is novel but the algorithm used is not so novel. It is simple and not so novel that the paper uses the greedy maximization algorithm to solve modality selection problem.

**Q5 Detailed Comments To The Authors:**

The experimental part of the article mainly uses two data sets for the binary classification task. If it is in other more complex tasks, is the proposed method effective?

**Q7 Justification For Your Score:**

See the above comments

**Q9 Complying With Reviewing Instructions:**

1: Yes.

---

### Decision · Program_Chairs · 2022-05-15

**Decision:**

Accept (Poster)

**Comment:**

Meta Review: The reviewers reach a consensus on the acceptance. The authors are encouraged to take all the comments into consideration and further improve the paper in the camera ready.